# Porous Thermoplastic Molded Regenerated Silk Crosslinked by the Addition of Citric Acid

**DOI:** 10.3390/ma16041535

**Published:** 2023-02-12

**Authors:** Alessio Bucciarelli, Nicola Vighi, Alessandra Maria Bossi, Brunella Grigolo, Devid Maniglio

**Affiliations:** 1Laboratorio RAMSES, IRCCS Istituto Ortopedico Rizzoli, Via di Barbiano 1/10, 40136 Bologna, Italy; 2Vetrodomus S.P.A., Via G. Bormioli 48, 25135 Brescia, Italy; 3Department of Biotechnology, University of Verona, Strada Le Grazie 15, 37134 Verona, Italy; 4Department of Industrial Engineering, BIOtech Research Center, University of Trento, Via delle Regole 101, 38123 Trento, Italy

**Keywords:** bone tissue engineering, bone scaffold, citric acid, silk resin, compression molding

## Abstract

Thermoplastic molded regenerated silk fibroin was proposed as a structural material in tissue engineering applications, mainly for application in bone. The protocol allows us to obtain a compact non-porous material with a compression modulus in the order of a Giga Pascal in dry conditions (and in the order of tens of MPa in wet conditions). This material is produced by compressing a lyophilized silk fibroin powder or sponge into a mold temperature higher than the glass transition temperature. The main purpose of the produced resin was the osteofixation and other structural applications in which the lack of porosity was not an issue. In this work, we introduced the use of citric acid in the thermoplastic molding protocol of silk fibroin to obtain porosity inside the structural material. The citric acid powder during the compression acted as a template for the pore formation. The mean pore diameter achieved by the addition of the higher amount of citric acid was around 5 μm. In addition, citric acid could effectively crosslink the silk fibroin chain, improving its mechanical strength. This effect was proved both by evaluating the compression modulus (the highest value recorded was 77 MPa in wet conditions) and by studying the spectra obtained by Fourier transform infrared spectroscopy. This protocol may be applied in the near future to the production of structural bone scaffolds.

## 1. Introduction

Silk fibroin (SF) is the internal structural protein of the silk bave and is responsible for its high mechanical strength. Silk fibroin has a semi-crystalline structure in which the β-sheet crystallites are rigid and mechanically resistant (highly ordered amino acid sequence, heavy chain, 300 kDa) whereas the amorphous phase (random coil secondary structure, light chain, 25 kDa) is flexible [1]. The unfolding of the β-structures during the protein dissolution and the possibility to tune their refolding allows us to change the material properties. This tunability, combined with the extreme versatility to be processed in mild conditions to obtain a wide range of architectures and its affinity to biological tissues, has made silk fibroin a perfect candidate for tissue engineering (TE) applications [1,2,3,4,5].

In bone tissue engineering (BTE), fibroin is used to develop biomimetic architectures with a porous structure [6,7,8,9,10]. Several strategies have been adopted, including but not limited to freeze-drying [11,12,13], salt-leaching [11,14,15], and foaming [16,17,18]. In almost all of them, mechanical stability is ensured by the folding of the secondary structure into β-sheets in a mechanism that is known as physical cross-linking [19,20,21,22]. Just a single method was reported that is based on the chemical cross-linking of a methacrylated silk fibroin (methacrylated silk, Sil-MA) [16,23]. The main issue with fibroin scaffolds is their mechanical strength, far from that of natural bone. This does not allow their use in structural applications in which a mechanical bearing is applied.

The only fibroin structures developed for BTE reported to have high mechanical performances are resins [24,25,26,27,28,29]. These resins have a compact non-porous structure suitable for structural applications but mainly intended for osteofixation due to the lack of a biomimetic architecture. Two main methods are used in the literature to produce them. The first is a sol–gel solid transition from the denatured protein in water or from a fibroin solution in hexafluoro isopropanol [30]. The solvent evaporation gradually allows the formation of a gel and then of compact solid material. However, this method requires a discrete amount of time to complete the evaporation and a second step to model the required object by subtractive manufacturing [30].

The latter method consists of a solid–solid transition obtained from a compression molding of a lyophilized fibroin sponge [28] or a fibroin powder [29] with low crystallinity. To set an appropriate process, several parameters need to be controlled. In particular, the temperature, the applied pressure profile, and the amount of water present in the starting material [28]. The principle of the solid–solid transition is the thermal reflow, a phenomenon that allows the molecules to rearrange under mechanical compression when the temperature is higher than the glass transition temperature (T_g_). The presence of water act as a plasticizer, reducing T_g_ and allowing an effective process even at low temperatures, down to 40 °C [28]. This method has been adopted to develop microstructured films [31] with control roughness and to produce an object in one step by compressing the starting material into a mold [29]. This resin was proved to reach compression modulus in the order of a gigapascal, however, when tested after a prolonged period in a pseudo-physiological condition, their mechanical resistance dramatically dropped. A chemical cross-linking was obtained by the addition of genipin to fibroin prior to the compression [32]. The cross-linking reaction triggered by the increase in temperature and the presence of water allowed us to obtain a chemically crosslinked resin with a compression modulus high enough to be used in structural applications even in wet environments such as the biological one [32].

The main disadvantage of all the previous protocols is the lack of porosity that does not allow to this resin to be applied as a structural scaffold. In this work, we propose a modified compression molding protocol that allows to contemporary crosslink the silk fibroin and obtain a porous structure. The addition of citric acid (CA) in powder ensured both this effect, the fraction dissolved enables the cross-linking reaction, while the other fraction acts as a pores template. CA has been previously used to crosslink starch [33], cellulose [34,35], chitosan [36,37], alginate [38], agarose [39], collagen [40], and gelatin [41]. In the case of silk, CA has been previously used on the raw fibers [42] and in several cases as a degumming agent to remove the external layer of sericin [43,44]. However, the cross-linking effect of CA on proteins was proved to occur at relatively mild conditions (low temperature in presence of humidity) and to act on the amine side groups [45].

The resin produced was proved porous by SEM imaging and image analysis, and the effect of the cross-linking was dependent on the amount of added CA. The resins have been mechanically tested by compression and the cross-linking reaction, namely the relative number of secondary structures, was evaluated by Fourier transform infrared spectroscopy (FTIR). To our knowledge, this is the first attempt to use CA as a cross-linking agent for silk fibroin and the first porous structural material produced with silk. A further complete biological evaluation will be needed to ensure the usability of this material for biomedical purposes.

## 2. Materials and Methods

### 2.1. Materials

Sodium carbonate monohydrate (Na_2_CO_3_ * H_2_O, Cat. 230952, ACS reagents, content >99.5%), lithium bromide (LiBr, Cat. 746479, Reagent Plus, content >99%), citric acid (CA, Cat. 251275, ACS reagents, content >99.5%) were bought by Sigma-Aldrich (St. Louis, MO, USA).

Silk cocoons were imported from Thailand (Chul Thai Silk Co., Phetchaban, Thailand).

### 2.2. Regenerated Silk Fibroin Preparation

The degumming process was completed following a procedure previously described [46,47] and shown in Figure 1A. Briefly, delaminated silk cocoons were placed in a bath of Na_2_CO_3_ aqueous solution, with a ratio of 1.1 g of salt for every liter of deionized water, at 98 °C for 90 min. In each degumming procedure, an amount of delaminated silk between 20 g and 30 g in weight was used (10 g per liter of solution). Then, to ensure the complete dissolution of sericin, the resulting silk was placed in a second bath of 0.4 g of Na_2_CO_3_ every liter of deionized water, at 98 °C for 90 min and under stirring. After this step, a different bath in deionized water at decreasing temperature was performed, trying to maintain the following steps of bath temperatures: 80 °C, 65 °C, 50 °C, 35 °C, and then a final bath at room temperature. The degummed silk fibroin fibers were then manually squeezed and manually separated, to remove any entangled parts. Finally, the material was wrapped in a double layer of paper and dried under the hood for at least 48 h. To ensure the complete sericin removal, the weight loss was checked to be between 25% and 30% as established by previous works [46,47].

The degummed silk fibroin was placed in a solution of LiBr 9.3 M with a ratio of 20% *w/v* placed in an oven (E028-230V-T, Binder, Bohemia, NY, USA) at 65 °C for 4 h, until complete dissolution. To remove the salt, the solution was dialyzed against deionized water using standard regenerated cellulose dialysis tubes (Spectra/Por, cutoff 3.4, 45 mm width), for 4 days. The resulting water solution was filtered to remove impurities and then checked in terms of pH and concentration. This last was determined by the spectroscopic method absorbance of the solution at 280 nm (nanodrop 1000, ThermoFisher Scientific, Waltham, MA, USA), using internal calibration of the instrument to determine the mg of fibroin for each mL.

### 2.3. Silk Fibroin Sponges

Knowing fibroin concentration in the dialyzed solution, typically around 40 mg/mL, the right amount of citric acid (CA) was added. Different concentrations of CA were selected based on the processability of the final compound and on the aim of obtaining an efficient cross-linking and pores formation. The objective was to add enough citric acid to show enhancement in mechanical properties given by the cross-linking effect and to leave a variable porosity based directly on the amount of citric acid added, with a further process of heat treatment, later described. Three main concentrations of citric acid were chosen, based on a preliminary evaluation, (from 0.25 to 2 in weight ratio CA/SF). These three concentrations, expressed in terms of ratio to the SF weight (CA/SF *w*/*w*), were 0.25, 0.50, or 0.75. Then, to completely remove water, we performed a freeze drying. Briefly, the samples were soaked in liquid nitrogen for 15 min to allow their freezing and then lyophilized for several days at −45 °C (Lio 5P, 5Pascal, Trezzano sul Naviglio, Italy). Due to the presence of CA, the samples were lyophilized in large Petri dish to increase the surface and speed up the process. After freeze-drying, the resulting samples were in a form of sponges and were stored in a dryer until their use.

### 2.4. Thermoplastic Molding

To produce the bulk resin, the freeze-dried material was compressed using a universal testing machine (858 Mini Bionix, MTS, Eden Prairie, MN, USA) as schematized in Figure 1A with an oven that allowed to set the temperature. The applied compression ramp is shown in Figure 1B. The compression cycle to produce bulk samples consisted of a linearly increasing load during 120 s until a maximum value of 10 kN, followed by a hold period of 30 s and a final period of 20 s where the load was fast reduced to zero. Compression was conducted in a plate–plate configuration, using a cylindrical mold (diameter of 6 mm, Figure 1B), properly designed to facilitate the removal of the specimen. Each sample was prepared starting from 200 mg of spongy material. Three different temperatures in the sintering process were used to study the activation of citric acid cross-linking: 40 °C, 80 °C, and 120 °C. Temperatures higher than 120 °C were discarded because they led to a compound with excessively low viscosity, determining the material flow through the mold’s slits.

The metal mold was pre-heated in the MTS furnace for two hours to permit the metal to reach the desired temperature. Because of the time needed to charge mold with the lyophilized material (operation made outside the oven) after every cycle of compression, a 5 min of heating step was performed to permit the mold to recover the processing temperature.

The presence of CA during the thermoplastic molding allows SF cross-linking. The hypothesized pathway is shown in Figure 1D and has been previously described in the literature [46]. Briefly, the amine groups of the lysine present in SF react with the carboxyl group of the CA to form an amide bond. This process repeated on the other carboxyl groups of CA generates a chemical cross-linking.

### 2.5. Thermal Treatment

In the attempt of obtaining surface and bulk porosity in the silk fibroin resin, a degradation of citric acid was induced. This concept took advantage of the presence of a large amount of citric acid entrapped in the fibroin structure and the capability to thermally release it. Since from the literature the thermal degradation of citric acid is reported starting at 153 °C, the sintered material was heated in oven (G-Therm, Fratelli Galli, Castelvetro di Modena, Italy) at 160 °C, following the temperature ramp described in Figure 1C. The temperature was maintained for 5 min at each thermal step, considering steading temperatures fixed at 80 °C, 100 °C, 120 °C, and 140 °C until reaching the final temperature of 160 °C. Samples were placed on a metallic grill over a glass Petri dish, to allow the release of citric acid below specimens.

Since after the heating step, for some samples, it was possible to observe melted citric acid on the surface, to improve its removal allowing porosity formation the samples were immersed in a beaker containing 100 mL of deionized water for 10 min, and then stored in a vacuum chamber for at least 48 h to allow the complete dehydration.

### 2.6. Morphological Analysis

Field emission scanning electron microscopy (FE-SEM Supra 60, Carl Zeiss, Jena, Germany) analysis was performed on fractured samples to observe cross-section porosity. To produce a fragile breakage, specimens were wrapped in aluminum foil and then immersed in liquid nitrogen for 3 min. Frozen specimens were then hit by a chisel using a hammer with a precise and rapid strike. Fragments of samples were stored in a vacuum chamber for 24 h, to ensure humidity removal. The cross-section surfaces were metallized by sputtering with a thin Pl/Pd layer. Micrography pictures were analyzed by FIJI software (National Institute of Health, v.1.53t), to obtain a quantitative result on pore size distribution by a segmentation with thresholding.

### 2.7. Mechanical Analysis

Mechanical characterization was performed in a pseudo-physiological condition, in compression with a universal testing machine (Instron 4502, Instron, Norwood, MA, USA). For each sample, three replicas were prepared and tested in wet conditions to mimic the real application environment by immersion of heat-treated samples in deionized water for 5 min or directly using samples after the salt leaching process. For each specimen diameter and width were measured using a caliber, to obtain precise data for stress/strain curves reconstruction. Samples were then mechanically tested with a plate-to-plate configuration using a 10 kN load cell. Compression test was made at a displacement rate of 1 mm/min as a standard parameter and test was conducted until breakage of sample or at least 1.5 mm of displacement, considering the material in a plastic deformation field.

Mechanical properties were then analyzed by comparing the three main samples with different citric acid ratios: 0.25 CA/SF (*w*/*w*), 0.5 CA/SF (*w*/*w*), and 0.75 CA/SF (*w*/*w*). Additionally, the study of the effect of sintering temperature on material elastic modulus was carried out.

### 2.8. Fourier Transform Infrared (FTIR) Analysis

Considering the secondary structure of silk fibroin as a protein, and its variations due to the cross-linking effect, a direct study on the secondary structure of the amine group of this protein was made. This study was completed on samples with different concentrations of citric acid, to observe the eventual variation of secondary structure responsible for the cross-linking phenomenon, in particular the β-sheet structure.

Every sample was dried in a vacuum chamber for 24 h before being tested with FTIR. The important evaluation was the study of the fundamental peaks, in particular the primary and secondary amide peaks, respectively, with center at 1620 cm^−1^ and 1538 cm^−1^. The infrared analysis was completed in total attenuated reflectance (ATR) mode (Spectrum ONE, Perkin Elmer, Waltham, MA, USA) measuring the transmittance spectra as mean of 16 scans in the range of primary amide 1590–1720 cm^−1^ with a resolution of 1 cm^−1^.

The analysis was performed as previously reported [48,49]. The primary amide peak was smoothed and then a Fourier self-deconvolution was applied (smoothing factor 0.3, gamma function 30). This allowed us to obtain well-defined peaks for each secondary structure contribution. Peak’s position was found by a second derivative. Then, the peaks were fitted by a Gaussian function, imposing the minimization of χ^2^. The assignment of the structure was completed following Table 1. The area of each pack was calculated and divided by the total area (sum of the area of the peak) to determine the relative percentage amount of the considered secondary structure.

## 3. Results

### 3.1. Citric Acid Addition and Lyophilization

As previously described, citric acid was added to obtain a cross-linking of the material and to obtain a certain degree of porosity. The addition of citric acid created some difficulties in the production steps of the standard process already established in our previous works [28,32]. The initial trials were conducted with an excess of citric acid compared to the weight of the regenerated silk, using a wide range of CA/SF ratio, ranging from 1 CA/SF to 4 CA/SF (as shown in Figure 2A, normal).

After lyophilization, an increase in the hardness was observed in the samples with CA content, but above 2 CA/SF the incomplete freeze-drying gave soft samples, clearly visible when pressure was manually applied (Figure 2A, compressed). It should be noticed that the sample with the highest amount of CA (4 CA/SF) was so soft that it was deformed during the extraction from the mold. This was attributed to the hygroscopicity of citric acid which retains water in the SF sponge, even during lyophilization. The case of 2 CA/SF is emblematic: while the external part was rigid and completely lyophilized, the sample’s core remained soft, a symptom of a non-complete freeze-dying, where CA contributed to keeping water in a liquid state. Sample 1 CA/SF was the sole to be completely lyophilized and, for this reason, it was considered as the maximum citric acid concentration allowable for completing the freeze-drying procedure. Once this limit was determined, the citric acid concentration considered for further characterization steps was set to 0.25 CA/SF, 0.5 CA/SF, and 1 CA/SF, which resulted in compact hard, sponge-like samples, as observed from the fracture surfaces (Figure 2B, cross).

### 3.2. Compression Molding

Despite freeze-drying process results, samples with high citric acid concentration were sintered to observe the effect of the natural crosslinker on the final bulk product. CA/SF ratios between 0.25 and 2 were evaluated, to analyze the further possibility of pores formation with citric acid degradation. Sintering process was performed at four different temperatures: room temperature (RT), 40 °C, 80 °C, and 120 °C. The increasing temperature was chosen to facilitate and improve the cross-linking capability of citric acid. In Figure 3A the produced samples are shown.

From the sintering process, it was possible to observe the typical change in color of fibroin, with increasing sintering temperature, from shiny white to transparent yellowish. The main problem in the use of citric acid during the sample production process was to reduce the viscosity of the material inside the mold, during the sintering step. The combined action of increasing temperature and citric acid concentration led to the reduction of viscosity of the material during sintering, with the loss of the cylindrical shape under compression. This resulted in the failure of the compression step, a problem that was encountered for samples with a citric acid ratio higher than 1 CA/SF.

As shown in Figure 3A, 1 CA/SF sample gave a low final shape fidelity of molded bulk materials for sintering temperatures higher than 40 °C. Instead, samples compressed at RT resulted to be difficult to control in terms of material converted to resin, and for this reason, they were excluded from the following phases.

### 3.3. Thermal Treatment

After the sintering process, a thermal treatment was performed to induce porosity formation. Samples were heated through a temperature ramp up to 160 °C to degrade citric acid, then removed with a deionized water bath to leave empty cavities. A set of thermally treated samples is reported in Figure 3B. Samples were produced by compression molding at 80 °C and increasing the amount of CA. A strong deformation of the samples was observed above 0.75 CA. This phenomenon was attributed to the strong citric acid gas expansion and its difficulty in escaping from the core of the molded cylinder. For this reason, samples with a CA ratio contentment above 0.75 were not further tested.

During the immersion in water, the samples floated (Figure 3C) as a result of the progressive cleaning from the residual CA. This was a proof of the formation of the pores and the decreasing in the sample density.

### 3.4. Morphological Analysis

Samples were broken to reveal the internal surface and analyze the pore distribution. The SEM micrographies are shown in Figure 4A while the comparison between the pore distributions in the box plots of Figure 4B, the descriptive statistic is summarized in Table 2. Regarding 0.25 CA/SF samples, the increase in sintering temperature did not lead to an evident variation in pore structure (as clearly visible from the micrographies). Fracture surface was characterized always by the presence of not-interconnected pores localized in few sites. The sample compressed at 40 °C evidenced interconnected porosity. The general structure for 0.25 CA/SF was characterized by a non-homogenous distribution of pores with a wide range in size. In addition, by the analysis of pores distribution, the mean equivalent diameter measured for each sintering temperature of 0.25 CA/SF sample was always below 5 µm, with the distribution width progressively broadening with the increase in the sintering temperature.

The 0.5 CA/SF samples showed an apparent increase in porosity with respect to 0.25 CA/SF samples and, in particular, for 40 °C treatment pores start to show an interconnected structure. This interconnection is apparently reduced increasing sintering temperatures to 80 °C, until 120 °C where it was completely lost. For both 0.25 CA/SF and 0.5 CA/SF samples, the microstructure obtained by compression molding at 120 °C was characterized by small, isolated pores, a fact that can be explained by the high mechanical resistance shown in the previous analysis. For this reason, the sintering temperature of 120 °C was considered inconvenient for the production process adopted. The quantitative analysis showed mean values of pores diameters lower than 5 µm, but with a large distribution reaching values up to 10 µm.

Analyzing micrographs for sample 0.75 CA/SF, pores appeared larger and better interconnected than previous ones. This enhancement of porosity could explain the lower mechanical properties, and, despite the higher interconnection, this citric acid concentration was difficult to be considered valid for material production. From the quantitative consideration, 0.75 CA/SF showed, for each sintering temperature, a further increase in the overall pore dimensions, reaching higher mean values, with pores size up 15 µm.

### 3.5. Mechanical Analysis

Citric acid crosslinked samples showed a visible compact structure considering the three main CA/SF ratios. Deformed samples were not measured, because they had a tendency to deteriorate even under manipulation.

Samples were analyzed after thermal treatment and for all the three sintering temperatures used. It is important to consider that every sample was tested in wet conditions, to mimic the final application conditions of the material. The mechanical tests were obtained in compression and load–displacement results were reported. Therefore, by the measure of specimens’ cylindrical dimensions, stress–strain curves were plotted, and elastic modulus was calculated (summarized in Table 3).

A general observation of the results shows a progressive increase in load with the displacement, and then an increase in stress with the material strain, and this is due to the continuous compaction of the material and the closure of the pores, initially present. Therefore, tests were carried out until an advanced deformation of the samples, in order to show this permanent collapse of pores. The initial part of the curves was interesting to extrapolate the elastic modulus, considering the characteristic behavior obtained before pores closure.

For 0.25 CA/SF samples, increasing the sintering temperature, no relevant effects were observed on the final mechanical properties, regarding the stress–strain curves (Figure 5A). In general, for all curves, an initial flat tract was considered as a settlement of the higher punch to the specimen surface. Samples were broken with a fragile behavior, with the formation of small debris, like powder. This brittleness was probably due to both the thermal treatment and the cross-linking.

For 0.5 CA/SF samples, the results were similar to those for the previous sample, with a general increase in mechanical properties for 120 °C of sintering temperature (Figure 5B), a fact that can be attributed to the higher amount of citric acid.

For 0.75 CA/SF samples, stress values are strongly reduced, and the bulk sintered material is not able to sustain the load after thermal treatment (Figure 5C). The high content of citric acid produced a material excessively fragile, so the sample was not considered reliable for the production of a porous scaffold.

To consider a quantitative measure of mechanical properties for these samples, the first part of the stress–strain curve was considered for the measure of the elastic modulus. In this tract of the curve, the material was elastically deformed, and no debris was formed. For each sample, there was not a continuous trend in the variation of the elastic modulus increasing the sintering temperature (Figure 5D). The higher values were obtained at 120 °C of sintering temperature, but data were remarkably scattered. Sintering at 40 °C showed a higher elastic modulus with respect to 80 °C, and for both, the results were more reliable in different specimens.

In addition, moving from 0.25 CA/SF to 0.5 CA/SF, the elastic modulus was increased for each sintering temperature used, but with a further increment of citric acid (to 0.75 CA/SF), the elastic modulus brutally decreased, as expected also by the trend of the stress–strain curves. For this reason, the excess of citric acid was neglected as a feasible solution for obtaining higher porosity.

### 3.6. Structural Analysis

In Figure 6A, the spectra of compressed molded silk fibroin (SF), and compression molded silk fibroin with the addition of citric acid (CA/SF 0.50) and citric acid (CA), are reported. All the samples were produced by compression molding at 80 °C. In the SF spectra, the peaks centered on 1617 cm^−1^, 1510 cm^−1^, and 1225 cm^−1^ were assigned to Amide I, Amide II, and Amide III, respectively. The two peaks detectable in CA at 1751 cm^−1^ and 1507 cm^−1^ indicate the C=O stretching of carboxylic acid. Those peaks disappeared in the CA/SF 0.50 spectra where a peak appeared centered at 1720 cm^−1^ indicating of the presence of carbonyl groups, which confirmed the protein cross-linking, as previously revealed in the literature [46]. Interestingly, the increment in the CA content produced an increase in the 1720 cm^−1^ peak, as highlighted by the arrow in Figure 6B indicating an increase in the cross-linking degree.

The Amide II peak was further deconvolved to reveal the different percentages of secondary structures before (Figure 6C) and after (Figure 6D) thermal treatment and compared to a sample prepared by compression molding of silk fibroin (SF) at 80 °C. The addition of CA increased the percentage amount of the crystalline β- phase at the expense of the random coil phase.

The increase in temperature from 40 °C to 80 °C decreased the effectiveness in the transition to β, while the increase to 120 °C was the more effective treatment. A higher number of crystalline structures was obtained by the compressions at 120 °C. After the thermal treatment (Figure 6B) the trend became less clear. This may be due to the complex effect of the CA decomposition and the contemporary trigger of the high temperature towards the fibroin crystallization.

## 4. Discussion

In this work, we studied the production of a silk resin by thermoplastic molding in which the addition of CA ensured both the formation of a porous structure and the cross-linking of the protein chains. The cross-linking effect of CA on proteins was previously studied [45], in the case of Silk, CA was used only to treat the degummed fibers in order to improve their strength [42]. To our knowledge, this is the first attempt to use CA with silk fibroin to develop a material suitable for tissue engineering.

Only a subset of the produced samples was chosen to investigate their properties. In this subset, the samples were sufficiently converted from the sponge to the resin, and they were not completely deformed by the thermal treatment. This included three CA/SF ratios 0.25, 0.5, and 0.75.

From the microstructural point of view while all previously developed protocols could not generate a porosity [24,25,26,27,28,29,32], the addition of CA was effective in generating pores which mean diameter was related to the amount of CA added. The higher the amount of CA the higher the mean diameter and the higher the number of pores generated. The higher amount of CA added (0.75 CA/SF) gave a structure that resemble a sponge but with thicker walls. It should be noticed that the porosity generated by our method was far from the porosity obtained with other methods (such as freeze-drying, foaming, and others) specifically developed to obtain a spongy SF architecture [11,12,13,14,15,16,17,18]. In fact, SF sponges usually have a wide pore dimensional distribution with diameters that reach hundreds of micrometers, while in our case we were able to obtain pores with diameters of tens of micrometers. However, this is to our knowledge the first successful attempt to produce a structural porous biopolymeric material.

The cross-linking effect was detected by the increase in the compression modulus moving from 0.25 CA/SF to 0.5 CA/SF while a further increase to 0.75 CA/SF ratio. This could be explained by the structural analysis. In fact, the increase to 0.75 CA/SF gave a particularly porous structure which was probably less resistant to the load bearing. The best result was achieved with a CA/SF ratio of 0.5 and a compression temperature of 120 °C that gave a compression modulus of around 80 MPa. Compared to the fully compact resins obtained in previous works this result was similar to the modulus obtained in the pseudo-physiological condition in the case of a silk resin crosslinked with genipin [32] (around 90 MPa) and higher than the modulus tested in wet conditions in not crosslinked SF resins [28,29] (in the 10–20 MPa range). This further confirmed the effect of CA on cross-linking the SF chains.

The cross-linking was confirmed also by the FTIR analysis in which the peak related to the presence of carbonyl groups, which intensity increased with the increase in CA content. It should be noticed that this increase was almost linear, moving from 0 CA/SF, 0.25 CA/SF to 0.50 CA/SF, but it reached a sort of plateau moving from 0.50 to 0.75. This may indicate that a complete cross-linking was reached at ratios under 0.75 CA/SF. The addition of CA increased the relative percentage number of β-sheets compared to the sample produced without CA. This effect was similar to one obtained by the dissolution of silk in formic acid the simpler organic acid and could be explained by the same mechanism, an interaction triggered by the presence of polar groups in both CA and SF that led to compact the SF molecules and generate intramolecular H-bonds [49,50,51]. Interestingly, the β-sheet content decreased moving from compression at 40 °C to 80 °C and then increased when the compression was performed at 120 °C. This may be explained by considering the complexity of the system in which chemical cross-linking bonding is involved. Probably at 40 °C a low degree of cross-linking still allows the β transition, while at 120 °C the thermal energy is so high to allow a transition even with a high degree of cross-linking. At 80 °C, the transition may be not inhibited by the presence of cross-linking and the lack of thermal energy.

In this work, we did not perform any biological evaluation, because of the wide literature present for both SF and CA, the sole two materials used in this study. However, in a follow-up study, we will investigate the biological performances of this material by in vitro screening.

## 5. Conclusions

The use of silk resins obtained by compression molding is generally not suitable for scaffolding due to the internal microstructure that is compact and non-porous. In this work, the addition of citric acid to the thermoplastic molding of silk achieved two purposes: produce a porosity inside to the compact structure and chemically crosslink the SF chain. This material had a compression modulus comparable to a previously developed crosslinked silk fibroin resin in wet conditions, reaching at best a modulus of around 80 MPa. The addition of porosity was the main achievement. By secondary electron microscopy, we analyzed the internal microstructure confirming that the undissolved citric acid acted as a template, leaving the pores once removed by thermal treatment followed by a water bath. The mean pore diameter reached was around 5 μm with a skewed distribution that reached 20 μm as the maximum diameter. Fourier transform infrared spectroscopy was used to confirm the effective cross-linking and to verify its mechanism. The best sample in terms of mechanical performance was obtained by compression at 120 °C with a citric acid/silk fibroin ratio of 0.50. This material may be used in the future as a structural bone scaffold in sites subjected to load bearing. A further complete biological evaluation and its correlation with the material swellability (the ability of this material to allow nutrient circulations through medium absorption) will be needed to understand the interaction of this material with the biological tissues and verify its usability as a biomedical device.

## Figures and Tables

**Figure 1 materials-16-01535-f001:**
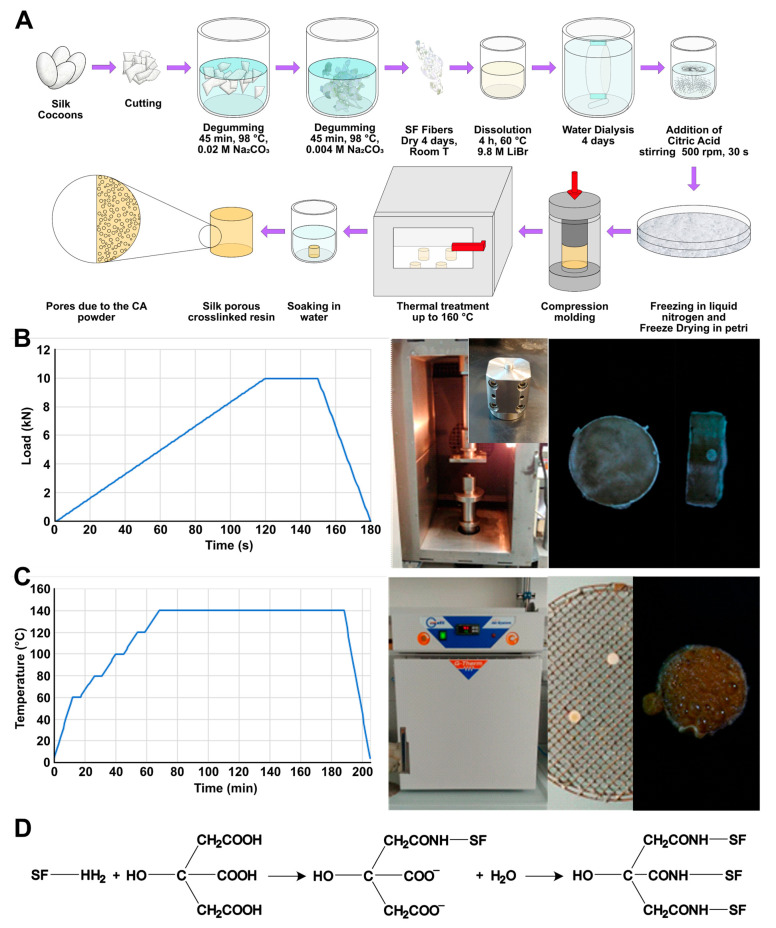
(**A**) Complete scheme of preparation. Briefly, the cocoons were cut and degummed by two Na_2_CO_3_ baths at different concentrations, the degummed fibers were then dissolved in a LiBr solution. The resulting denatured fibroin solution was dialyzed to remove the salt. Based on the concentration of fibroin in solution, a calculated amount of citric acid was added and rapidly mixed. The solution was then frozen in liquid nitrogen and freeze-dried to produce a sponge. The sponge was compressed into a mold at a determined temperature. The compression-molded fibroin resin was then thermally treated to allow the fusion of CA and then placed in water to remove its presence. The resulting object was porous. (**B**) Compression phase. The compression was performed following a ramp, reaching 10 kN in 120 s, holding the pressure for 30 s, and releasing it in 30 s. The compression was performed in a universal testing machine with an oven that ensure the possibility of maintaining a constant temperature. The mold was designed specifically to easily extract the compressed samples. An example of a prepared sample is shown in figure. (**C**) The thermal treatment was conducted in oven following the reported temperature ramp up to 160 °C to allow the CA melting and release. The samples were placed on a grinder, to let the fused CA dripping. After thermal treatment the sample resulted to be wet, confirming the effectiveness of the procedure. (**D**) A possible pathway for SF cross-linking according to the literature SF amine residues reacts with the carboxyl group to form amide bond. This process repeated on the multiple carboxyl groups of CA allows the chemical cross-linking.

**Figure 2 materials-16-01535-f002:**
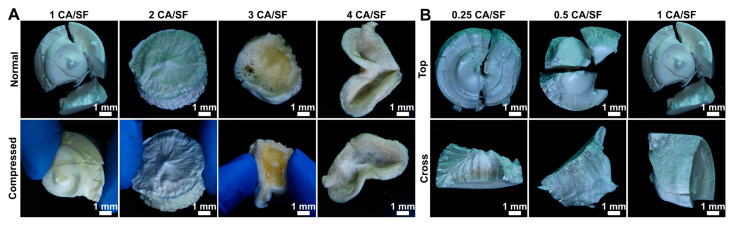
(**A**) Sponges with addition of citric acid (CA) with a CA/SF ratio (*w*/*w*) ranging from 1 to 4. Only the sample with the lower amount of CA was completely lyophilized. The samples resulted to be less deformable under manual compression with the increasing of CA up to 2 CA/SF, above the hygroscopicity of the salt made freeze-drying not feasible. It should be noticed that also the core of the 2 CA/SF sample resulted to be still not dry. (**B**) The samples used for the compression molding were produced with CA/SF ratios ranging from 0.25 to 1. This last was chosen as the upper limit from the initial evaluation.

**Figure 3 materials-16-01535-f003:**
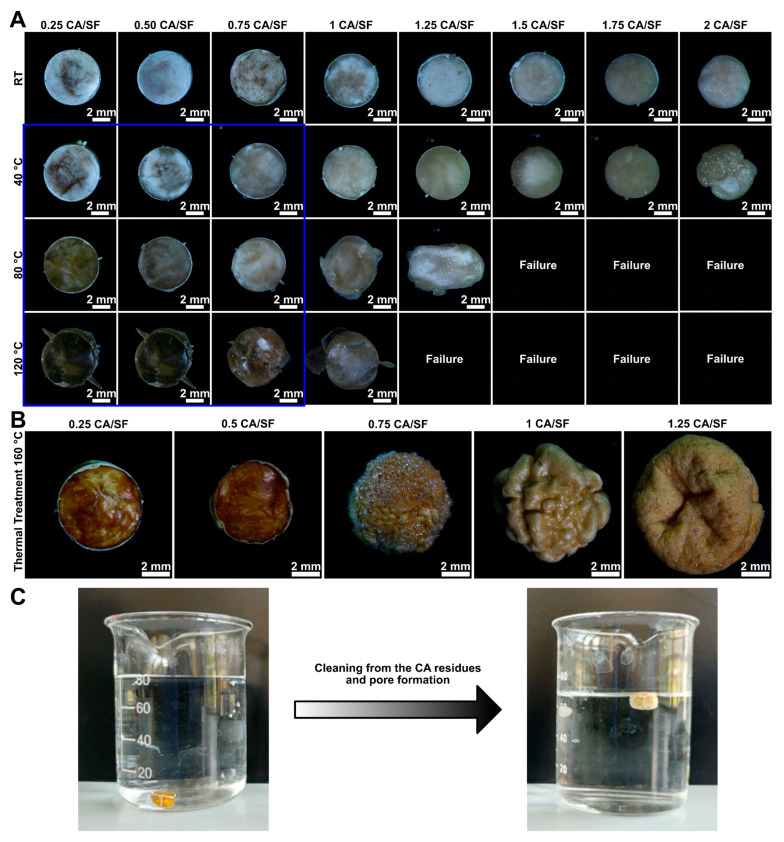
(**A**) Fibroin samples from the compression molding step. The samples indicated with failure were deformed completely during the compression. The shape was satisfactory in a wide range of conditions. However, we tested only the samples that did not lose their shape during the following thermal treatment and excluded the samples produced at room temperature (RT) that did not ensure an effective solid-solid transition. The tested samples are enclosed in the blue rectangle. (**B**) Thermal treatment (up to 160 °C) of samples produced with compression molding at 80 °C. Samples with a CA equal to or above 1 CA/SF dramatically changed their shape. For this reason, only samples with lower amounts of CA were tested in the following phases. (**C**) Step of CA removal in deionized water. Samples floated as an effect of the CA removal.

**Figure 4 materials-16-01535-f004:**
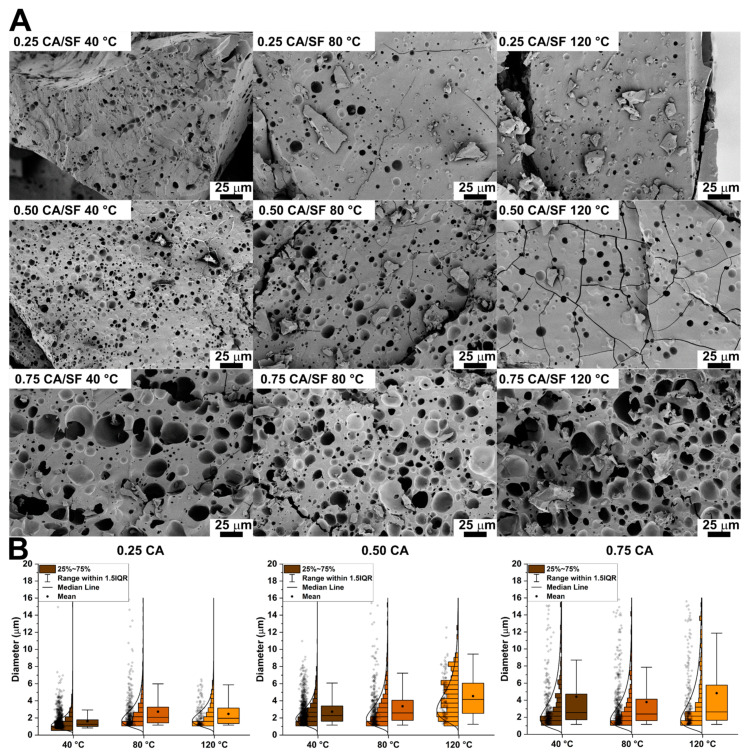
(**A**) SEM images collected from the cross-section of the samples. The increase in the amount of CA increased the porosity while the increase in temperature did not influence the porosity. (**B**) Box plot comparing the pore diameter distributions with different CA/SF ratios obtained at different temperatures but the same CA/SF ratio. The box plots are reported with the data points, the distribution histogram, and the log-normal distribution used to fit them.

**Figure 5 materials-16-01535-f005:**
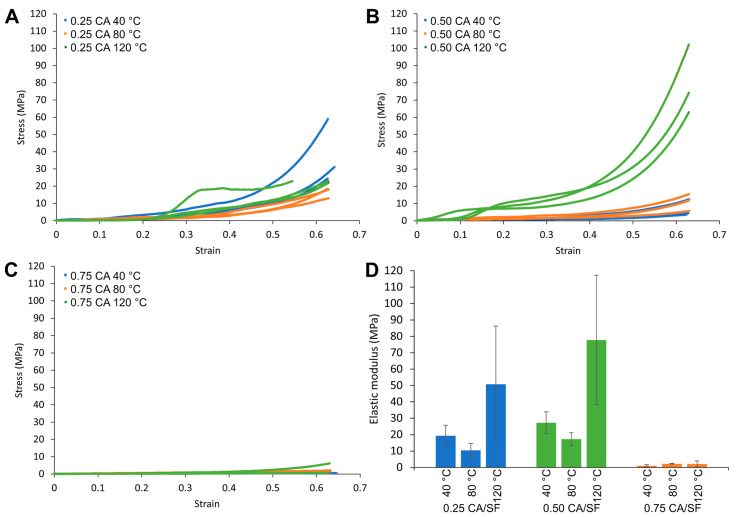
Stress–strain curves for samples compressed at different temperatures with the CA/SF ratios of (**A**) 0.25 (**B**) 0.50 and (**C**) 0.75. All the samples were tested in a pseudo-physiologic condition (after water soaking at 37 °C. (**D**) Summary results in term of compression modulus. An increment of the compression modulus was recorded by the increment in CA/SF ratio from 0.25 to 0.50 however a further increment in CA (CA/SF 0.75) content strongly decreased the compression modulus.

**Figure 6 materials-16-01535-f006:**
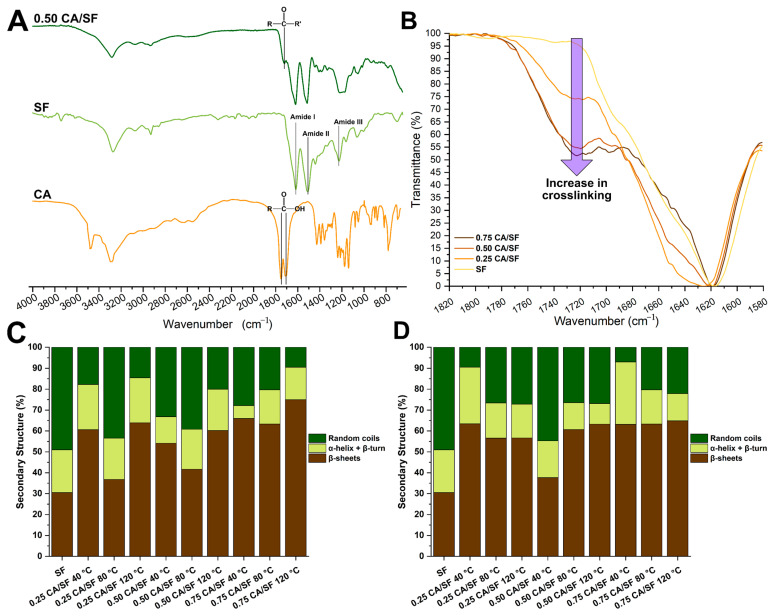
(**A**) FTIR spectra of citric acid (CA), a sample obtained by compression molding of lyophilized silk fibroin (SF-CA), and a sample in which CA was added prior to compression molding (SF_0.50CA-CM). by comparing the spectra, it was possible to spot the formation of a peak indicating the presence of carboxyl groups when CA was added to fibroin, which confirms the protein cross-linking. (**B**) The intensity of carboxyl peak increased (as indicated by the purple arrow) with the increase in the CA amount indicating a higher degree of cross-linking. (**C**) Relative amount in percentage of secondary structures before the thermal treatment. (**D**) Relative amount in percentage of secondary structures after the thermal treatment. The presence of CA increased the crystalline structure compared to the compressed sample without CA (SF). A small effect of the thermal treatment on the relative percentage was detected.

**Table 1 materials-16-01535-t001:** Peaks assignment for the quantification of the percentage of secondary structures.

Secondary Structure	Wavenumber Range (cm^−1^)
Side chain	1597–1609
β-sheets	1610–1625
β-sheets	1626–1635
Random coils	1636–1655
α-helix	1656–1662
β-turns	1663–1696
β-sheets	1697–1703

**Table 2 materials-16-01535-t002:** Descriptive statistics of the pore diameter distributions for the different citric acid/silk fibroin (CA/SF) rations and temperature (T). The mean diameter (Mean), the standard deviation (StD), the distribution skewness (Skew) and kurtosis (Kurt), the minimum diameter (Min), the first quartile (Q1), the median, the third quartile (Q3), the maximum diameter (Min) and the interquartile range (IQR) were extrapolated. All of them resulted to be skewed to the lower values, however with the increasing of the CA/SF ratio, the distribution become wider (IQR increases).

CA/SF	T	Mean	StD	Skew	Kurt	Min	Q1	Median	Q3	Max	IQR
(*w*/*w*)	(°C)	(μm)	(μm)	(μm)	(μm)	(μm)	(μm)	(μm)	(μm)	(μm)	(μm)
0.25	40	2.47	1.37	1.17	0.49	1.17	1.35	1.96	3.16	6.51	1.81
0.25	80	2.73	2.08	2.95	11.64	1.17	1.43	2.08	3.27	15.81	1.84
0.25	120	1.62	1.07	3.75	29.06	0.83	1.01	1.26	1.78	14.95	0.77
0.50	40	2.74	1.54	1.57	2.62	1.17	1.62	2.27	3.40	10.97	1.79
0.50	80	3.37	2.52	2.36	7.14	1.17	1.72	2.59	4.07	17.29	2.35
0.50	120	4.56	2.34	0.68	0.10	1.26	2.54	4.17	6.06	12.56	3.52
0.75	40	4.43	4.64	2.43	5.91	1.17	1.72	2.55	4.73	26.39	3.01
0.75	80	3.79	3.71	2.63	7.75	1.17	1.62	2.38	4.13	22.69	2.51
0.75	120	4.83	4.93	2.12	5.07	1.17	1.65	2.63	5.78	30.29	4.13

**Table 3 materials-16-01535-t003:** Mean and standard deviation of the compression modulus from the produced samples, the condition that gave us the best result was a CA/CF ratio of 0.5 compressed at 120 °C. Sample with 0.75 CA/SF ratio was structurally compromised by the presence of pores.

CA/SF	T (°C)	Mean E (MPa)	StD E (MPa)
0.25	40	19.23	6.52
0.25	80	10.39	4.32
0.25	120	50.61	35.57
0.5	40	27.14	6.84
0.5	80	17.20	4.07
0.5	120	77.47	39.61
0.75	40	0.99	0.94
0.75	80	2.19	0.41
0.75	120	2.15	2.06

## Data Availability

Data are available from the authors upon reasonable request.

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
