# Peer review of "Porous Thermoplastic Molded Regenerated Silk Crosslinked by the Addition of Citric Acid"

_materials, 2023, doi:10.3390/ma16041535_

Round 1
Reviewer 1 Report
The research paper entitled: Porous thermoplastic molded regenerated silk crosslinked by the addition of citric acid shows an interesting research idea. This article could be interesting to the readers of the journal Materials. Despite that, some should be addressed. A major revision is necessary. The following issues should be addressed:
1. In the abstract you use abbreviations that are not defined but also the full terms for units. Better to unify that and not to use abbreviations in the abstract, as a better reading flow is possible.
2. Space sign between a value and a unit are mostly missing throughout the whole manuscript and also in scale bars in figure 4A. This is also valid for figures. For example, as in figure 1: 98°C and 160 °C.
3. Check the manuscript carefully again. Sometimes commas are missing. For example: line 426. But there are more.
4. Scale bars in figure 2 and 3 are missing. Please add them. Readers should get an idea about the real size of the samples shown.
5. In chapter 2: Add information about all used chemicals (purity, manufacturer, city, country), as this is completely missing.
6. Add the correct information about all used devices (type, manufacturer, city, country).
7. Space signs between the text and the reference are missing. Sometimes also in the common text or figures (figure 4A, for example) space signs are missing. Please add them.
8. Figure 1B, 1C, 4B, 5A, 5B and 5C: The axis labels are too small and not good readable as figure 5D is better presented.
9. In table 1, the unit is in this brackets [], the units in table 2 are shown without brackets, as the figures are using these brackets (). Please unify the manuscript to one type of brackets.
10. The diagrams in Figure 4B are too tiny. Please arrange figure 4 to solve this issue.
11. Sometime, the degree sign is used not correctly (for example: line 421, line 436, …). Use the correct one: Alt + 0176 Moreover, mostly the space sign is missing before the degree sign. Please add them throughout the whole manuscript.
12. Table 2 caption should mention all parameters shown full term and abbreviation.
13. Figure 6B: The meaning of the pink arrow should be added in the figure caption.
14. In line 338, the authors writing about swelling of the samples. The swelling rate for the samples should be determined, provided in the manuscript and discussed.
15. What about the roughness, wettability and surface energy of the samples? These parameters should be measured and provided in the manuscript and discussed, as they are important parameters for the cell adhesion and even important for the planned in vitro study, mentioned by the authors. Moreover, the authors wrote: “In this work we did not perform any biological evaluation, because of the wide literature present for both SF and CA the sole two materials used in this study.” – I agree with that point. But, to be able to compare the other literature with the samples studied here, the samples must be characterized by the important parameter affecting the cell cultivation in in vitro and in vivo experiments. This is important, as here the SF where modified with CA.
16. The conclusion is too less. Please rewrite the conclusion as the main results are not good presented. As for the abstract, it is suggested to avoid to use abbreviations.
Author Response
Reviewer 1
The research paper entitled “Porous thermoplastic molded regenerated silk crosslinked by the addition of citric acid” shows an interesting research idea. This article could be interesting to the readers of the journal Materials. Despite that, some should be addressed. A major revision is necessary. The following issues should be addressed:
We thank the reviewer for his/her comment, that allows us to substantially improve our manuscript.
- In the abstract you use abbreviations that are not defined but also the full terms for units. Better to unify that and not to use abbreviations in the abstract, as a better reading flow is possible.
We thank the reviewer for his/her comment. In the abstract, as requested, the abbreviations have been substituted by the full names.
- Space sign between a value and a unit are mostly missing throughout the whole manuscript and also in scale bars in figure 4A. This is also valid for figures. For example, as in figure 1: 98°C and 160 °C.
We thank the reviewer for his/her comment. The space have been inserted between numbers and unit of measurements.
- Check the manuscript carefully again. Sometimes commas are missing. For example: line 426. But there are more.
As suggested by the reviewer, the manuscript has been checked and corrected.
- Scale bars in Figure 2 and 3 are missing. Please add them. Readers should get an idea about the real size of the samples shown.
We apologize for the lack of scale bars. In the revised version of the manuscript, scale bars were added.
- In chapter 2: Add information about all used chemicals (purity, manufacturer, city, country), as this is completely missing.
We thank the reviewer for his/her comment, as suggested a paragraph named “Materials” have been added describing the used chemicals.
- Add the correct information about all used devices (type, manufacturer, city, country).
As suggested, all the information regarding the used instruments have been added to the manuscript.
- Space signs between the text and the reference are missing. Sometimes also in the common text or figures (figure 4A, for example) space signs are missing. Please add them.
We thank the reviewer, as suggested space signs have been added.
- Figure 1B, 1C, 4B, 5A, 5B and 5C: The axis labels are too small and not good readable as figure 5D is better presented.
We thank the reviewer for his/her comment. Figure 1 and Figure 5 have been changed to make axis readable.
- In table 1, the unit is in this brackets [], the units in table 2 are shown without brackets, as the figures are using these brackets (). Please unify the manuscript to one type of brackets.
We thank the reviewer for his/her suggestion. One type of bracket has been chosen and unified throughout the manuscript.
- The diagrams in Figure 4B are too tiny. Please arrange figure 4 to solve this issue.
We thank the reviewer for his/her comment. As suggested, Figure 4 has been rearranged to make the diagrams visible.
- Sometime, the degree sign is used not correctly (for example: line 421, line 436, …). Use the correct one: Alt + 0176 Moreover, mostly the space sign is missing before the degree sign. Please add them throughout the whole manuscript.
We thank the reviewer for his comment. The incorrect symbol has been substituted by the correct one.
- Table 2 caption should mention all parameters shown full term and abbreviation.
As requested, the caption has been modified as follows: “Descriptive statistic of the pore diameter distributions for the different citric acid/silk fibroin (CA/SF) ratio and temperature (T). The mean diameter (Mean), the standard deviation (StD), the distribution skewness (Skew) and kurtosis (Kurt), the minimum diameter (Min), the first quartile (Q1), the median, the third quartile (Q3), the maximum diameter (Max) and the interquartile range (IQR) were extrapolated. All of them resulted to be skewed to the lower values, however with the increasing of the CA/SF ratio, the distribution become wider (IQR increases)”.
- Figure 6B: The meaning of the pink arrow should be added in the figure caption.
We thank the reviewer for his/her comment. As suggested the meaning of the pink arrow has been added to the caption.
- In line 338, the authors writing about swelling of the samples. The swelling rate for the samples should be determined, provided in the manuscript and discussed.
We thank the reviewer for his/her comment. We apologize for the incorrect use of the term swelling; the material gets deformed by the effect of the fusion of citric acid and not by the effect of water. The term swell has been substituted by deformed.
We do agree that the swelling is an important property, and it will be discussed in a further work correlated to the biological properties. Swelling is important to diffuse nutrients inside the material and allowing the growth and proliferation of cells. This has been added in the conclusion as follows: “A further complete biological evaluation and its correlation with the material swellability (the ability of this material to allows the nutrient circulations through medium absorption) will be needed to understand the interaction of this material with the biological tissues and verify its usability as biomedical device”.
- What about the roughness, wettability and surface energy of the samples? These parameters should be measured and provided in the manuscript and discussed, as they are important parameters for the cell adhesion and even important for the planned in-vitro study, mentioned by the authors. Moreover, the authors wrote: “In this work we did not perform any biological evaluation, because of the wide literature present for both SF and CA the sole two materials used in this study.” – I agrees with that point. But to be able to compare the other literature with the samples studied here, the samples must be characterized by the important parameter affecting the cell cultivation in in-vitro and in-vivo experiments. This is important, as here the SF were modified with CA.
We thank the reviewer for his comment; we do agree that the surface properties should be evaluated as parameters to predict the biological properties, but in the case of films or patches, where the cell culture is 2D. It should be noticed that they are rarely determined in case of fibroin made materials, and never in the case of this type of constructs [1–8]. Scaffolds are commonly not evaluated in term of surface properties because of their 3D architecture, which make the evaluation of the surface properties not straightforward. In addition, the literature is not coherent on the role of wettability and surface energy on the biological outcome, in fact, in a work conducted on a large library of materials the measurement of the contact angle failed in predicting microbial and stem cell attachment [9]. This is also the case of silk fibroin where, as other biomaterials, the presence of cell attachment sites may not strongly relate with the physical property of the surface.
The roughness instead is something that for this material can be tuned according to the surface that compresses the sponge. In fact, SF when compressed at high pressure undergoes a phenomenon known as thermal reflow in which the molecules can “flow” to reconfigure the material. This has been used to imprint SF films with microstructures. The roughness in this case is not a property to be considered unchangeable. The possibility to tune the roughness has been inserted in the introduction as follows: “This method has been adopted to develop microstructured films [10] with control roughness and to produce object in one step by compressing the starting material into a mold [11]”.
- The conclusion is too less. Please rewrite the conclusion as the main results are not good presented. As for the abstract, it is suggested to avoid using abbreviations.
We thank the reviewer for his/her comment. In the conclusions, as requested, the abbreviations have been substituted by the full names. In addition, the conclusion has been extended.
References
- Tuan, H.A.; Hirai, S.; Tamada, Y.; Akioka, S. Preparation of silk resins by hot pressing Bombyx mori and Eri silk powders. Mater. Sci. Eng. C 2019, 97, 431–437, doi:10.1016/J.MSEC.2018.12.060.
- Anh Tuan, H.; Hirai, S.; Inoue, S.; A. H. Mohammed, A.; Akioka, S.; Ngo Trinh, T. Fabrication of Silk Resin with High Bending Properties by Hot-Pressing and Subsequent Hot-Rolling. Materials (Basel). 2020, 13, 2716, doi:10.3390/ma13122716.
- Kaneko, A.; Tamada, Y.; Hirai, S.; Kuzuya, T.; Hashimoto, T. Characterization of a silk-resinified compact fabricated using a pulse-energizing sintering device. Macromol. Mater. Eng. 2012, 297, 272–278, doi:10.1002/mame.201100112.
- Tirta Nindhia, T.G.; Koyoshi, Y.; Kaneko, A.; Sawada, H.; Ohta, M.; Hirai, S.; Uo, M. Hydroxyapatite-silk functionally graded material by pulse electric current sintering. Trends Biomater. Artif. Organs 2008, 22, 25–29.
- Tao, Y.; Xu, W.; Yan, Y.; Wu, H. Structure and properties of composites compression-molded from silk fibroin powder and waterborne polyurethane. Polym. Adv. Technol. 2012, 23, 639–644, doi:10.1002/pat.1938.
- Bucciarelli, A.; Chiera, S.; Quaranta, A.; Yadavalli, V.K.; Motta, A.; Maniglio, D. A Thermal‐Reflow‐Based Low‐Temperature, High‐Pressure Sintering of Lyophilized Silk Fibroin for the Fast Fabrication of Biosubstrates. Adv. Funct. Mater. 2019, 29, 1901134, doi:10.1002/adfm.201901134.
- Yu, W.; Kuzuya, T.; Hirai, S.; Tamada, Y.; Sawada, K.; Iwasa, T. Preparation of Ag nanoparticle dispersed silk fibroin compact. Appl. Surf. Sci. 2012, 262, 212–217, doi:10.1016/j.apsusc.2012.05.084.
- Bucciarelli, A.; Janigro, V.; Yang, Y.; Fredi, G.; Pegoretti, A.; Motta, A.; Maniglio, D. A genipin crosslinked silk fibroin monolith by compression molding with recovering mechanical properties in physiological conditions. Cell Reports Phys. Sci. 2021, 0, doi:10.1016/J.XCRP.2021.100605.
- Alexander, M.R.; Williams, P. Water contact angle is not a good predictor of biological responses to materials. Biointerphases 2017, 12, 02C201, doi:10.1116/1.4989843.
- Brenckle, M.A.; Tao, H.; Kim, S.; Paquette, M.; Kaplan, D.L.; Omenetto, F.G. Protein-protein nanoimprinting of silk fibroin films. Adv. Mater. 2013, 25, 2409–2414, doi:10.1002/adma.201204678.
- Guo, C.; Li, C.; Vu, H. V.; Hanna, P.; Lechtig, A.; Qiu, Y.; Mu, X.; Ling, S.; Nazarian, A.; Lin, S.; et al. Thermoplastic Molding of Regenerated Silk. Nat. Mater. 2020, 19, 102, doi:10.1038/S41563-019-0560-8.

Reviewer 2 Report
The manuscript “Porous thermoplastic molded generated silk crosslinked by the addition of citric acid” is an interesting new procedure for generating porous biomaterials. The methodology for the production of the porous material is innovative, and it is well explained. However, the characterization of the materials is a little bit poor; the porosity is characterized by SEM, which is well done. The mechanical properties of porous materials by compression, and the crosslinking process is proved by FTIR, where in my opinion the degradation of CA is clear, but the crosslinking grade is proved qualitatively. Finally, the discussion and conclusion are well done; this manuscript can be published in materials. But in my opinion, to improve this manuscript, this biomaterial should be characterized by other thermal analysis, some of them very common, as DSC or DMA. In those techniques, the glass transition can be observed, or even monitor the crosslinking process during the thermal treatment at different temperatures in case CA degradation is not relevant; TGA could analyze this point in advance.
Other minor points to correct are:
Page 2, line 79. Alginate appeared two times.
Page 12, line 416. Authors should cite some reference when they mentioned “as previously revealed in literature”
Author Response
Reviewer 2
The manuscript “Porous thermoplastic molded generated silk crosslinked by the addition of citric acid” is an interesting new procedure for generating porous biomaterials. The methodology for the production of the porous material is innovative, and it is well explained. However, the characterization of the materials is a little bit poor; the porosity is characterized by SEM, which is well done. The mechanical properties of porous materials by compression, and the crosslinking process is proved by FTIR, where in my opinion the degradation of CA is clear, but the crosslinking grade is proved qualitatively. Finally, the discussion and conclusion are well done; this manuscript can be published in materials.
We thank the reviewer for his/her comment.
- But in my opinion, to improve this manuscript, this biomaterial should be characterized by other thermal analysis, some of them very common, as DSC or DMA. In those techniques, the glass transition can be observed, or even monitor the crosslinking process during the thermal treatment at different temperatures in case CA degradation is not relevant; TGA could analyze this point in advance.
We thank the reviewer for his/her comment. DSC and DMA were performed in our previous works on the same topic. The glass transition temperature depends on the amount of water in silk fibroin. And in the case of a crosslinked complex biological material, as SF crosslinked with CA, the glass transition temperature is not easily determined by thermal analysis. The choice to not characterized this material by thermal analysis is dictated by its possible usage. In fact, the purpose is to use it at a low temperature (37 degree) in a biological system. The degradation as well as the biological in-vitro response will be analyzed in a further work. To clarify this point, the following sentence have been added to the conclusion: “A further complete biological evaluation and its correlation with the material swellability (the ability of this material to allow the nutrient circulations through medium absorption) will be needed to understand the interaction of this material with the biological tissues and verify its usability as biomedical device”.
- Other minor points to correct are: Page 2, line 79. Alginate appeared two times. Page 12, line 416. Authors should cite some reference when they mentioned “as previously revealed in literature”.
We thank the reviewer for his/her comment. Minor points have been corrected as suggested.

Reviewer 3 Report
The authors develop a porous structural material with a compression modulus in the order of a Giga Pascal by compressing lyophilized sponge of silk fibroin powders. The use of citric acid in the thermoplastic molding process not only increases the mechanical properties of the material as a crosslinking agent, but also acts as a template for the pore formation after thermal treatment to form porous structural materials. This protocol is very instructive for tissue engineering applications, such as structural bone scaffold. The results are interesting and the presentation is clear. The paper is recommended for publication in Materials. There are several comments suggested below.
1. The crosslinking between citric acid and silk fibroin are the key innovation in this paper. The FTIR characterization suggest that there are peaks of citric acid in the composite materials. How is citric acid linked to silk fibroin?
2. The authors claim that the porous structural material has a compression modulus in the order of a Giga Pascal. Is the compression modulus measured before or after the dissolution of residual citric acid?
3. A work on porous materials, e.g. Adv. Mater. 2021, 33, 2102362, and a work on silk fibroin, e.g. Adv. Mater. Interfaces DOI:10.1002/admi.202201962, are suggested.
4. Why are the content of β-sheets and elastic modulus of porous materials treated at 80 °C lower than those treated at 40 °C and 120 °C?
5. There are some typos in the text and figure, for example, “ad oven” in the figure caption and “0.25 CA” in Figure 5B and C.
Author Response
Reviewer 3
The authors develop a porous structural material with a compression modulus in the order of a Giga Pascal by compressing lyophilized sponge of silk fibroin powders. The use of citric acid in the thermoplastic molding process not only increases the mechanical properties of the material as a crosslinking agent, but also acts as a template for the pore formation after thermal treatment to form porous structural materials. This protocol is very instructive for tissue engineering applications, such as structural bone scaffold. The results are interesting, and the presentation is clear. The paper is recommended for publication in Materials. There are several comments suggested below.
We thank the reviewer for his/her comment, that allows us to substantially improve our manuscript.
- The crosslinking between citric acid and silk fibroin are the key innovation in this paper. The FTIR characterization suggest that there are peaks of citric acid in the composite materials. How is citric acid linked to silk fibroin?
The link between CA and silk fibroin is probably related to the lysine amino acids which have amine side groups, as previously described [1]. The hypothesized scheme of reaction has been reported in Figure 2D. A brief explanation has also been inserted in the manuscript as follows: “The presence of CA during the thermoplastic molding allows the SF crosslinking. The hypothesized pathway is shown in Figure 1D, and has been previously described in the literature [1]. Briefly, the amine groups of the lysine present in SF reacts with the carboxyl group of the CA to form an amide bond. This process repeated on the other carboxyl groups of CA generate a chemical crosslinking.”.
- The authors claim that the porous structural material has a compression modulus in the order of a Giga Pascal. Is the compression modulus measured before or after the dissolution of residual citric acid?
We apologize with the reviewer for the lack of clarity. The porous structure does not reach the compression modulus of gigapascal as the bulk silk resin in dry conditions does. In wet condition, SF resin strongly decreased their compression modulus in the order of tens megapascal. The crosslinking in the porous structure allows to obtain values similar to the values obtained in case of genipin crosslinked resin. To clarify this point, Table 3 with the numerical values has been added and, in the discussion, the following part was inserted: “The best result was achieved with a CA/SF ratio of 0.5 and a compression temperature of 120 °C that gave a compression modulus around 80 MPa. Compared to the fully compact resins obtained in previous works this result was similar to the modulus obtained in pseudo-physiological condition in case of a silk resin crosslinked with genipin [2] (around 90 MPa) and higher than the modulus tested in wet conditions in not crosslinked SF resins [3,4] (in the 10 MPa-20 MPa range). This further confirmed the effect of CA in crosslinking the SF chains”.
- A work on porous materials, e.g. Adv. Mater. 2021, 33, 2102362, and a work on silk fibroin, e.g. Adv. Mater. Interfaces DOI:10.1002/admi.202201962, are suggested.
We thank the reviewer for his/her suggestion. The references have been both added to the manuscript.
- Why are the content of β-sheets and elastic modulus of porous materials treated at 80 °C lower than those treated at 40 °C and 120 °C?
We thank the reviewer for his/her comment. This is an interesting aspect; we are not sure about the reason, but we can hypothesize that in this case the presence of CA may play an important role. As we know from the literature, the increase in the temperature generally increases the β content, however in our system, the chemical crosslinking may inhibit this phenomenon. Probably with a compression at 40°C the degree of crosslinking is not so high, and the transition is possible, while at 120°C the kinetic energy is so high to allow a transition even if the chemical crosslinking is present. 80°C may be the temperature range in which the transition is not possible. To clarify this point, the following comment has been added to the discussion: “Interestingly, the β-sheet content decreased moving from a compression at 40◦C to 80 °C and then increased when the compression was performed at 120 °C. This may be explained considering the complexity of the system in which chemical crosslinking bonding is involved. Probably at 40 °C a low degree of crosslinking still allows the β transition, while at 120 °C the thermal energy is so high to allow a transition even with a high degree of crosslinking. At 80 °C the transition may be not inhibited by the presence of crosslinking and the lack of thermal energy”.
- There are some typos in the text and figure, for example, “ad oven” in the figure caption and “0.25 CA” in Figure 5B and C.
We thank the reviewer for his/her comment. A complete grammar check has been done and the minor errors corrected.
References
- Xu, H.; Shen, L.; Xu, L.; Yang, Y. Low-temperature crosslinking of proteins using non-toxic citric acid in neutral aqueous medium: Mechanism and kinetic study. Ind. Crops Prod. 2015, 74, 234–240, doi:10.1016/j.indcrop.2015.05.010.
- Bucciarelli, A.; Janigro, V.; Yang, Y.; Fredi, G.; Pegoretti, A.; Motta, A.; Maniglio, D. A genipin crosslinked silk fibroin monolith by compression molding with recovering mechanical properties in physiological conditions. Cell Reports Phys. Sci. 2021, 0, doi:10.1016/J.XCRP.2021.100605.
- Bucciarelli, A.; Chiera, S.; Quaranta, A.; Yadavalli, V.K.; Motta, A.; Maniglio, D. A Thermal‐Reflow‐Based Low‐Temperature, High‐Pressure Sintering of Lyophilized Silk Fibroin for the Fast Fabrication of Biosubstrates. Adv. Funct. Mater. 2019, 29, 1901134, doi:10.1002/adfm.201901134.
- Guo, C.; Li, C.; Vu, H. V.; Hanna, P.; Lechtig, A.; Qiu, Y.; Mu, X.; Ling, S.; Nazarian, A.; Lin, S.; et al. Thermoplastic Molding of Regenerated Silk. Nat. Mater. 2020, 19, 102, doi:10.1038/S41563-019-0560-8.

Reviewer 4 Report
In the abstract
Both two abbreviations (SF and CA) should be written as a full name at first mention
The most promising results obtained should be mentioned in the abstract as numerical values
In introduction part
what does the BTE abbreviation mean
correct two main methods are use in literature to used.
The alginate was mentioned two times, while another crosslinked materials in the cited papers …. kindly, review the cited papers (31,32 and 33)
31- potato starch/chitosan
32- chitosan
33- alginate/polyvinyl alcohol
The introduction contains only one 2022 reference cited. kindly add some recent studies to the introduction
In materials part
all materials used in this work should be mentioned with their origin, main chemical and physical properties
expand the scale in Fig. A,B and C
Author Response
Reviewer 3
We thank the reviewer for his/her contribution that allowed us to improve our manuscript.
In the abstract
- Both two abbreviations (SF and CA) should be written as a full name at first mention
Both abbreviations have been inserted in the introduction at their first occurrence.
- The most promising results obtained should be mentioned in the abstract as numerical values.
Numerical values have been included in the abstract, as suggested by the reviewer. The following part has been inserted in the abstract: “The mean pore diameter achieved by the addition of the higher amount of citric acid was around 5 μm. In addition, the citric acid was able to effectively crosslink the silk fibroin chain improving the mechanical strength. This effect was proven both by evaluating the compression modulus (the highest value obtained was 77 MPa in wet conditions) and by studying the spectra obtained by Fourier transform infrared spectroscopy”.
In introduction part
- what does the BTE abbreviation mean
We apologize with the reviewer; the full name has been added at the first occurrence. BTE stays for Bone Tissue Engineering.
- correct two main methods are use in literature to used.
We thank the reviewer. The sentence has been corrected.
- The alginate was mentioned two times, while another crosslinked material in the cited paper, kindly, review the cited papers (31,32 and 33), 31-potatostarch/chitosan, 32-chitosan, 33-alginate/polyvinyl alcohol.
We thank the reviewer for his comment. The citations have been modified and referred to the proper biopolymer.
- The introduction contains only one 2022 reference cited. kindly add some recent studies to the introduction
We do agree that in the introduction only one recent paper has been cited, however citric acid is not so commonly used as crosslinker and there are only few papers on silk resins. It should be noticed that most of the papers cited in the reference were published between 2018-2022, which is considerable as recent literature. To fulfill the reviewer's request, we added two relevant references published the last year.
In materials part
- All materials used in this work should be mentioned with their origin, main chemical, and physical properties.
We thank the reviewer for his/her comment. An additional paragraph entitled “Materials” has been added to the manuscript.
- expand the scale in Fig. A, B and C
We thank the reviewer for his suggestion. The scale has been expanded.

Round 2
Reviewer 1 Report
The authors of the manuscript entitled “Porous thermoplastic molded regenerated silk crosslinked by the addition of citric acid” addressed and discussed my comments in an appropriate way. After revision, the quality of the manuscript raised significantly.
Author Response
We thank the reviewer for his/her comment.
Reviewer 2 Report
The revision has improved the manuscript, clarifying the small points that I commented in my first revision. Then, this version can be published in Materials.
Author Response

(The authors gave the same response as above.)
